# Efficient ethylene purification by a robust ethane-trapping porous organic cage

Kongzhao Su [1,2,4], Wenjing Wang [1,4], Shunfu Du[1,3], Chunqing Ji[1,2] & Daqiang Yuan [1,2 ✉]

The removal of ethane ($C_2H_6$) from its analogous ethylene ($C_2H_4$) is of paramount importance in the petrochemical industry, but highly challenging due to their similar physicochemical properties. The use of emerging porous organic cage (POC) materials for $C_2H_6/C_2H_4$ separation is still in its infancy. Here, we report the benchmark example of a truncated octahedral calix[4]resorcinarene-based POC adsorbent (CPOC-301), preferring to adsorb $C_2H_6$ than $C_2H_4$, and thus can be used as a robust absorbent to directly separate high-purity $C_2H_4$ from the $C_2H_6/C_2H_4$ mixture. Molecular modelling studies suggest the exceptional $C_2H_6$ selectivity is due to the suitable resorcin[4]arene cavities in CPOC-301, which form more multiple C–H···π hydrogen bonds with $C_2H_6$ than with $C_2H_4$ guests. This work provides a fresh avenue to utilize POC materials for highly selective separation of industrially important hydrocarbons.

[1] State Key Laboratory of Structural Chemistry, Fujian Institute of Research on the Structure of Matter, Chinese Academy of Sciences, Fuzhou, Fujian, China. [2] University of the Chinese Academy of Sciences, Beijing, China. [3] College of Chemistry, Fuzhou University, Fuzhou, China. [4] These authors contributed equally: Kongzhao Su, Wenjing Wang. ✉email: ydq@fjirsm.ac.cn

Ethylene ($C_2H_4$), the largest feedstock in petrochemical industries with a global production capacity of more than 170 million tons in 2018, exceeds any other chemical feedstock molecules for its wide application in polyethylene manufacture[1]. The industrial separation of $C_2H_4$ from ethane ($C_2H_6$) is typically through cryogenic distillation at high pressure and very low temperature, using very high towers, because they have similar sizes and volatilities[2]. Such a process is highly energy consuming, and thus exploring of other effective $C_2H_4$ separation methods at ambient conditions is highly demanded. Selective adsorption by porous materials for producing high-purity $C_2H_4$ has been determined to be one of the most desired methods for its low energy consumption[3,4]. Of particular recent interest in this regard is using $C_2H_6$-selective porous materials enriched with nonpolar/inert surfaces (e.g., introducing aromatic or aliphatic entities), because they prefer to capture more polarizable $C_2H_6$, and thus can directly afford high-purity $C_2H_4$ in a single adsorption step, avoiding an additional high-energy-consuming desorption step[5,6]. So far, most of the developed $C_2H_6$-selective adsorbents are metal-containing materials[6–10]. However, the polar metal centers in these materials can interact with unsaturated $C_2H_4$ molecules via strong π-complexation interactions, and thus weaken their $C_2H_6/C_2H_4$ separation abilities[11]. Therefore, developing other $C_2H_6$-selective adsorbents such as metal-free organic porous materials is highly required[12,13], because they cannot only avoid the abovementioned problem, but also are easier to construct inert surfaces. However, the research of such materials is still in its initial stage[14,15].

Porous organic cages (POCs), as a type of porous materials, are intrinsically porous given their hollow cavities[16–21]. The discrete inherent nature of POCs makes them possessing distinct benefits in solution processing, regeneration as well as post-synthesis modification[21–24]. Following the first elegant research reported in 2009 by Cooper et al.[25], the number of POCs with different shapes, sizes and properties has increased substantially[26–36], but robust POCs with high surface areas are still few, which highly hinder their practical use in gas storage as well as separation. Over the past decade, although Brunauer–Emmett–Teller (BET) surface areas of POCs have increased from the initial 624 to 3758 $m^2 g^{-1}$, most of POCs have BET values <1000 $m^2/g$[37,38]. As for binary gas mixture separation, Cooper et al. first reported that imine-linked tetrahedral POC (CC3) exhibits selectivity of 20.4 for noble gas Xe/Kr separations at low concentrations in 2014[39]. Two years later, they showed that CC3 could also separate sulfur hexafluoride ($SF_6$), a much more potent greenhouse gas than carbon dioxide, from nitrogen ($N_2$) with the highest $SF_6/N_2$ selectivity reported for any material under ambient conditions[40]. The selectivity for the abovementioned two cases arises from the precise size matching between Xe and $SF_6$ with organic cage

cavity than Kr and $N_2$, respectively. In 2019, their team utilized post modification synthesis method to alter internal cavities of CC3 to produce hybrid cocrystal material (6ET-RCC3-R/CC3-S), which are excellent quantum sieves for hydrogen isotope separation with excellent deuterium/hydrogen selectivity (8.0)[41]. In the same year, Zhang's and Zaworotko's groups presented a soft imide-based POC (NKPOC-1) with gate opening behavior and could efficiently separate binary or ternary C3 hydrocarbon mixtures[42]. Up to now, POC adsorbents have only been limited to separate the above gas components by mixed gas experiments, and thus there is still much room to explore POC materials for real gas separation applications, especially for industrially important gases.

Calix[4]resorcinarenes, a subset of calixarenes, were derived from acid-catalyzed condensation between resorcinol and various kinds of aldehydes[43–45]. They possess electron-rich π cavities and eight upper-rim phenolic groups, and are effective hosts for inclusion of various guests ranging from small gases to large organic molecules[45–49]. Notably, their upper rims can be easily functionalized, making them good molecular building blocks for constructing self-assembled cages as well as porous polymers[50–56]. Very recently, our group employed predesigned concave-shaped tetraformylresorcin[4]arene (RC4ACHO) as secondary building blocks and different diamines as linkers to systematically design and synthesize several porous POCs with structural diversity from [2 + 4] dimeric lanterns, [3 + 6] trimeric triangular prisms, to [6 + 12] hexameric octahedra[57]. However, utilizing calix[4]resorcinarene-based POCs as solid adsorbents for gas separation applications remains unexploited. Herein, we present that the robust and highly porous [6 + 12] octahedral calix[4]resorcinarene-based POC (CPOC-301) is an excellent $C_2H_6$-selective material, and can be used as a robust absorbent to directly afford high-purity $C_2H_4$ from $C_2H_6/C_2H_4$ mixture.

## Results

**Crystal structure and characterization of CPOC-301.** As shown in Fig. 1a, CPOC-301 is successfully prepared via self-assembly of RC4ACHO (1 equiv.) and p-phenylenediamine (2 equiv.) under mild condition. Single-crystal X-ray diffraction reveals that CPOC-301 has a truncated octahedron structure, with eight trigonal ports having edge length reaching about 12 Å, and a large cavity with inner diameter as well as volume that respectively reach 16.8 Å and 4270 $Å^3$. The solid-state packing of CPOC-301 suggests that it possesses a one-dimensional channel, with a diameter of ~7 Å, viewed from [001] direction (Fig. 1b). Notably, quantitative tautomerization of imines in CPOC-301 to their keto-enamine forms has been observed from $^1H$ NMR and FT-IR spectroscopy (Supplementary Figs. 1–3). Such a transformation

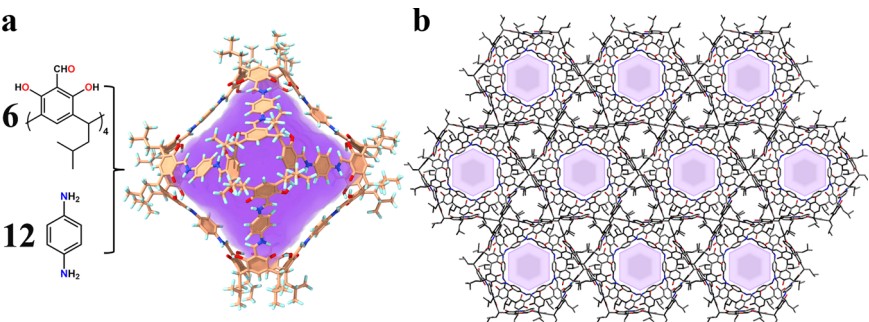

**Fig. 1 Schematic illustration for assembly of CPOC-301. a** The X-ray crystal structure of CPOC-301. **b** The solid-state molecular packing of CPOC-301 viewed from [001] direction, where H atoms are omitted for clarity. color codes: phenyl ring; orange, carbon; gray, oxygen; red, nitrogen; blue, and hydrogen; light turquoise.

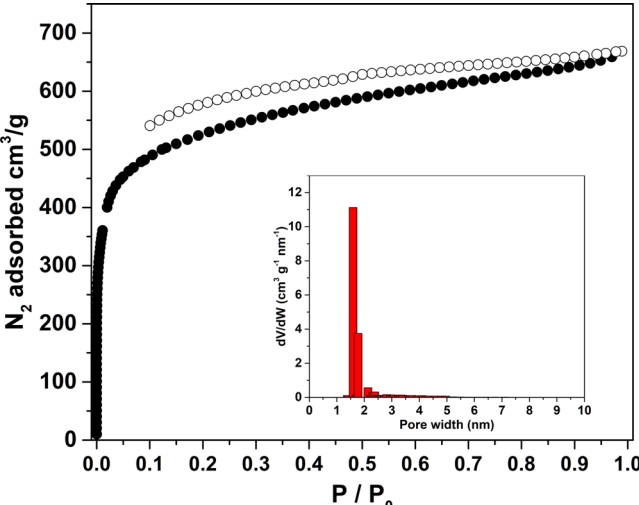

**Fig. 2 Porosity measurements.** $N_2$ gas sorption isotherm at 77 K for CPOC-301, inset: the calculated PSD of CPOC-301.

makes CPOC-301 more stable even when it is stored in air for over half a year, as has been confirmed by FT-IR spectra as well as $^1$H NMR studies (Supplementary Figs. 3 and 4). Thermal gravimetric analysis (TGA) suggests that CPOC-301 exhibits thermal stability up to 300 °C under nitrogen atmosphere (Supplementary Fig. 5). PXRD reveals that desolvated CPOC-301 retains its crystal packing (Supplementary Fig. 6), which is very uncommon in large POC systems.

**Gas adsorption and separation performances.** The channel-type structure of CPOC-301 can potentially lead to interesting solid-state gas sorption properties. The permanent porosity of the activated CPOC-301 was confirmed by $N_2$ gas sorption experiments at 77 K (Fig. 2). Its $N_2$ adsorption isotherm shows a typical type I curve with a small fraction of a type IV adsorption behavior. The maximum $N_2$ adsorption is 670 $cm^3\,g^{-1}$, and the calculated BET of CPOC-301 is up to 1962 $m^2\,g^{-1}$. The pore volume as well as the micropore volume of CPOC-301 respectively reach 1.03 and 0.46 $cm^3\,g^{-1}$. The pore size distribution (PSD) profile exhibits a relatively sharp distribution of micropores from 1.59 to 1.77 nm (Fig. 2 inset), consistent with the cavity size of CPOC-301 in the crystal structure.

The high surface area and high density of nonpolar calix[4]resorcinarene cavities in CPOC-301, which may favor the preferential adsorption toward more polarizable $C_2H_6$ over $C_2H_4$, prompted us to investigate the sorption of these gases. Single-component adsorption isotherms of CPOC-301 for $C_2H_6$ and $C_2H_4$ were measured at 273, 283, and 293 K at 1 bar. Notably, CPOC-301 exhibits preferential adsorption of $C_2H_6$ (87 $cm^3\,g^{-1}$) over $C_2H_4$ (75 $cm^3\,g^{-1}$) at 293 K (Fig. 3a), and also the other two temperatures (Supplementary Figs. 7 and 8). The corresponding isosteric heat of adsorption ($Q_{st}$) at zero coverage for $C_2H_6$ and $C_2H_4$ was calculated to be 32.4 and 24.2 $kJ\,mol^{-1}$, respectively (Fig. 3b and Supplementary Figs. 9 and 10). Such a difference suggests that the host-guest interactions between CPOC-301 and $C_2H_6$ are much stronger than that of $C_2H_4$. Notably, the $C_2H_6$ uptake capacity of CPOC-301 at 293 K and 1 bar exceeds most of the reported porous organic materials[14] and $C_2H_6$-selective MOF materials[5].

Motivated by the high uptake capacity and $C_2H_6$-selective behavior of CPOC-301, the ideal adsorbed solution theory (IAST) is used to assess its separation selectivity for $C_2H_6/C_2H_4$ (50:50) (Supplementary Figs. 11–13). IAST calculation results demonstrate that $C_2H_6/C_2H_4$ selectivity range is from 1.3 to 1.4 at 293 K.

Compared to other $C_2H_6$-selective porous organic molecular materials, the selectivity value of $C_2H_6$ over $C_2H_4$ for CPOC-301 is comparable to [4 + 6] boronic ester cage (1.29) reported by Mastalerz et al.[58], but lower than the recently reported hydrogen-bonded organic frameworks including HOF-76 (2.0)[14] and ZJU-HOF-1 (2.25) reported by Li and Chen (Supplementary Fig. 14)[15]. To evaluate the actual separation performance of CPOC-301, the experimental breakthrough studies were conducted in a packed column of the activated CPOC-301 sample under an equimolar $C_2H_6/C_2H_4$ mixture at ambient conditions. The breakthrough curves depicted in Fig. 3c prove that CPOC-301 can efficiently realize the complete separation of $C_2H_4$ from $C_2H_6/C_2H_4$ mixtures. Notably, $C_2H_4$ gas breaks through the adsorption bed first to produce an outflow of pure gas containing no detectable $C_2H_6$. Conversely, $C_2H_6$ gas breaks through column following a substantial time-lapse, because the $C_2H_6$ molecule is more preferentially adsorbed in CPOC-301 than $C_2H_4$. From dynamic breakthrough experiment, the calculated separation factor for an equimolar mixture of $C_2H_6/C_2H_4$ was 1.3, consistent with the predicted IAST result. For practical industrial applications, the ideal adsorbent should also have good recycling performance. We performed multiple $C_2H_6/C_2H_4$ mixed-gas dynamic break-through experiments under similar operating conditions. The separation performance of $C_2H_6/C_2H_4$ does not obviously change within seven continuous cycles (Fig. 3d). Besides, NMR and PXRD data after breakthrough experiments in addition to $N_2$ gas adsorption and PXRD of the sample after being exposed to air and soaked in water (Supplementary Figs. 15–19), indicate that CPOC-301 is robust enough to be a promising candidate for $C_2H_4$ purification.

**Separation mechanism.** To further understand the role of CPOC-301 in the mechanism of selective $C_2H_6/C_2H_4$ adsorption, modeling studies based on an efficient conformer search algorithm (CREST)[59,60] and first-principles dispersion-corrected density functional theory (DFT-D) calculations were performed[14,15]. The primary binding sites for $C_2H_6$ and $C_2H_4$ molecules were found to be located at the calix[4]resorcinarene cavities. The lowest-energy gas binding configurations are displayed in Fig. 4a, b. For more clarity, only one adsorbed gas molecule at calix[4]resorcinarene cavity site is shown, because the remaining five sites within CPOC-301 are crystallographically identical. Notably, we believed that C2 hydrocarbon guests occupy the six primary binding sites of CPOC-301, because one CPOC-301 molecule was found to adsorb about 22.5 $C_2H_6$ and 19.5 $C_2H_4$ molecules as calculated from adsorption isotherms at 293 K. The corresponding calculated static binding energies ($\Delta E$, $\Delta E = E_{POC+gas} - E_{POC} - Egas$) of $C_2H_6$ and $C_2H_4$ are around −42.7 and −41.2 $kJ\,mol^{-1}$, respectively. Such binding energy variation suggests the stronger host-guest interactions between $C_2H_6$ and calix[4]resorcinarene cavity than that of $C_2H_4$, consistent with our aforementioned experimental observation. The higher binding energy of $C_2H_6$ is mainly ascribed to nonplanar $C_2H_6$ molecule sterically "matches" better to the electron-rich cavity of calix[4]resorcinarene than planar $C_2H_4$ molecule[14,15], because C–H···π bonds of guest $C_2H_6$ is more numerous than that of $C_2H_4$ (12 for $C_2H_6$ and 7 for $C_2H_4$; and specific H···π distances are listed in Supplementary Fig. 20). Moreover, the Hirshfeld surface was performed to further reveal the intermolecular contacts present within the corresponding simulated crystal structures[61,62]. Notably, the strong intermolecular interactions between C2 hydrocarbons and calix[4]resorcinarene cavity are indicated as bright red spots on the Hirshfeld surface (Supplementary Fig. 21), which are close to the abovementioned interaction regions of C–H···π bonds between the C2 guests and calix[4]resorcinarene host. Moreover, $C_2H_6$ and $C_2H_4$ gas sorption as well as $C_2H_6/C_2H_4$ mixture breakthrough

**Fig. 3 C$_2$H$_6$/C$_2$H$_4$ separation performances. a** Experimental C$_2$H$_6$ and C$_2$H$_4$ adsorption isotherms of CPOC-301 at 293 K. **b** Isosteric heat of adsorption plots for the adsorption of C$_2$H$_6$ and C$_2$H$_4$ by CPOC-301. **c** Experimental breakthrough curves for equimolar mixture of C$_2$H$_6$/C$_2$H$_4$ at 298 K and 1 bar over a packed bed of CPOC-301. **d** The recyclability of CPOC-301 under multiple mixed gas column breakthrough tests.

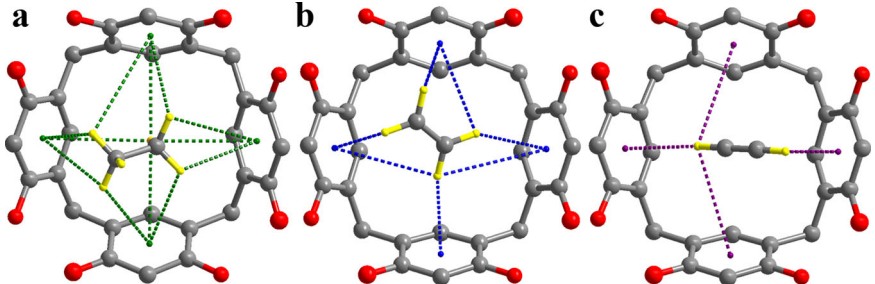

**Fig. 4 Mechanism study.** A comparison of preferential **a** C$_2$H$_6$, **b** C$_2$H$_4$, and **c** C$_2$H$_2$ adsorption sites and close C–H···π interactions within the calix[4] resorcinarene cavities observed by DFT-D calculations. Carbon is gray, oxygen red, and hydrogen yellow. Dashed bonds highlight C–H···π interactions.

experiments by other types of calix[4]resorcinarene-based POCs comprising one hexameric octahedron with functional methyl groups (CPOC-301-Me) and a trimeric triangular prism (CPOC-201) were measured to confirm the aforementioned host-guest interaction results[57]. Both of them show the same behavior in C$_2$H$_6$ and C$_2$H$_4$ gas sorption and separation compared to CPOC-301 (Supplementary Figs. 22–27). To further authenticate our result, host-guest interactions between CPOC-301 and C$_2$H$_2$ (acetylene),

another guest molecule of the C2 hydrocarbons family, has also been simulated (Fig. 4c). The calculated ΔE value of C$_2$H$_2$ is −36.7 kJ mol$^{-1}$, indicating that its host-guest interaction is weakest in the C2 hydrocarbons. This is consistent with the experimental $Q_{st}$ result with value of 20.2 kJ/mol for C$_2$H$_2$ (Supplementary Figs. 28 and 29), and has been confirmed by C$_2$H$_6$/C$_2$H$_2$ IAST calculation (Supplementary Fig. 30), C$_2$H$_4$/C$_2$H$_2$ and C$_2$H$_6$/C$_2$H$_4$/C$_2$H$_2$ breakthrough experiments (Fig. 5 and Supplementary Fig. 31).

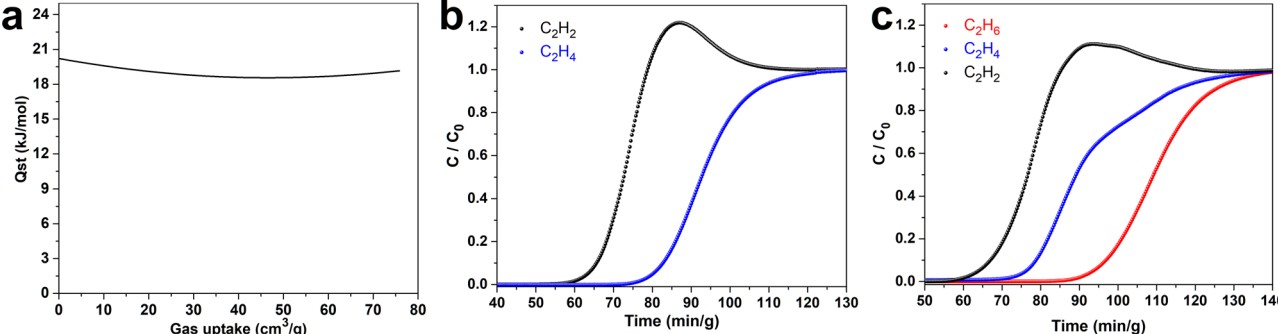

**Fig. 5 $C_2H_4/C_2H_2$ and $C_2H_6/C_2H_4/C_2H_2$ separation performances. a** Isosteric heat of adsorption plots for the adsorption of $C_2H_2$ by CPOC-301. **b** The experimental column breakthrough curve of CPOC-301 with an equimolar $C_2H_2/C_2H_4$ mixture. **c** The experimental column breakthrough curve of CPOC-301 with $C_2H_6/C_2H_4/C_2H_2$ mixture.

## Discussion

We report a pioneering work using shape-persistent amine-linked CPOC-301 as POC adsorbent for $C_2H_4$ purification. Specifically, the highly porous CPOC-301 can efficiently trap $C_2H_6$ from $C_2H_6/C_2H_4$ mixture and therefore directly produce high-purity $C_2H_4$. The preferential interactions with $C_2H_6$ over $C_2H_4$ in CPOC-301 are because $C_2H_6$ form more multiple C–H···π hydrogen bonds with resorcin[4]arene cavities than $C_2H_4$ guests as indicated by results of DFT-D calculations. This finding could shed some light on the design and synthesis of POCs based on supramolecular cavitands as "porous additives" in column and membrane separation applications for industrially important gases in the future. Efforts to explore these possibilities are ongoing.

## Method

**Characterization.** Under ambient temperature, proton nuclear magnetic resonance (1H NMR) data were collected by means of a Burker AVANCE 400 (400 MHz) spectrometer. Fourier-transformed infrared spectroscopy (FT-IR) spectra were taken on a Magna 750 FT-IR spectrometer utilizing KBr pellets in the 400–4000 $cm^{-1}$ region. High-resolution electrospray ionization mass spectrometry (ESI-TOF-MS) was collected using a Bruker MaXis™ 4G instrument. The TGA was collected at 10 °C/min ramp rate in dynamic $N_2$ flow within a temperature range of 30–900 °C by a NETZSCH STA 449C thermal analyzer. The pattern of powder X-ray diffraction (PXRD) was collected on a Rigaku Mini 600 X-ray diffractometer for $CuK_{\alpha}$ radiation ($\lambda = 0.154$ Å), with 0.5°/min scan speed of and 0.02° in 2θ step size.

**Synthetic procedure.** All reagents and solvents were supplied by Sinopharm Chemical Reagent Co., Ltd with analytical grade, and utilized without further purification. CPOC-301 was synthesized as follows[57]: 162 mg (0.20 mmol) RC4ACHO[63] and 43 mg (0.4 mmol) p-phenylenediamine were added into 5 mL nitrobenzene and 15 mL $CHCl_3$. After sealing the mixture in a 48 mL pressure vial, heating to 65 °C with stirring for 2 days, and cooling down naturally. Red block single crystals of CPOC-301 with ~78% yield were obtained by slow methanol vapor diffusion into the above mixture. 1H NMR (400 MHz, $CDCl_3$, 298 K): δ 1.04 (d, 144H), 1.60 (m, 24H), 2.10 (t, 48H), 4.65 (t, 24H), 7.37 (s, 24H), 7.40 (s, 48H), 9.18 (s, 24H), 10.38 (s, 24H), 16.21 (s, 24H). p.p.m. ESI-TOF-MS calculated for CPOC-301, $C_{360}H_{384}N_{24}O_{48}$ [M-2H]$^{2-}$ 2905.4178, found 2905.3963.

**Gas adsorption measurements.** Automatic volumetric adsorption equipment (Micromeritics, ASAP 2020) was utilized to conduct all gas adsorption-desorption measurements of CPOC-301. The data of PSD were obtained from the $N_2$ sorption isotherm at 77 K based on the DFT model in the Micromeritics ASAP 2020 software package (assuming cylinder pore geometry). Before measurements, samples' degassing was accomplished at 100 °C for 10 h under dynamic vacuum (below 10 μmHg) for 10 h for removing the adsorbed impurities. The calculated pore volume and micropore volume are based on built-in software of ASAP 2020 physisorption analyzer. The isosteric heat of sorption for C2 hydrocarbons was regarded as a function of gas uptake using compared the adsorption isotherms at 273, 283, and 293 K. After data being modeled with a virial-type expression comprising $a_i$ and $b_i$ parameters (Eq. (1)), the heat of adsorption ($Q_{st}$) is determined through fitting parameters by means of Eq. (2), in which $P$ refers to the pressure, $N$ refers to the adsorbed amount, $T$ refers to the temperature, $R$ refers to universal gas constant, while $m$ and $n$ determine the number of terms required to describe the isotherm adequately. The parameters were obtained from fitting of C2 hydrocarbons adsorption isotherms fitted with $R^2 > 0.999$. To assess $C_2H_6/C_2H_4$ separation

performance, IAST of Myers and Prausnitz[64] and pure component isotherm fits by dual-site Langmuir–Freundlich equation were employed for calculating molar loadings in the mixture for specific partial pressures of bulk gas phase (Eq. (3)), where $N$ refers to molar loading of species (mmol $g^{-1}$), A refers to saturation capacity of species (mmol $g^{-1}$), B refers to Langmuir constant (kPa$^{-c}$), C refers to Freundlich constant and P refers to bulk gas phase pressure of species (kPa). The adsorption selectivity based on IAST for $C_2H_6/C_2H_4$ mixture is identified using Eq. (4):

$$\ln p = \ln N + \frac{1}{T}\sum_{i=0}^{m} a_i N^i + \sum_{i=0}^{n} b_i N^i \tag{1}$$

$$Q_{st} = -R\sum_{i=0}^{m} a_i N^i \tag{2}$$

$$N = A_1 \frac{B_1 \times P^{C_1}}{1 + B_1 \times P^{C_1}} + A_2 \frac{B_2 \times P^{C_2}}{1 + B_2 \times P^{C_2}} \tag{3}$$

$$S_{A/B} = \frac{x_A y_B}{x_B y_A} \tag{4}$$

**Column breakthrough experiments.** The mixed-gas breakthrough separation experiment was performed through a home-built setup equipped with a mass spectrometer (Pfeiffer GSD320). For instance, in a typically conducted breakthrough experiment for $C_2H_6/C_2H_4/He$ (10:10:80, v/v/v), $C_2H_4/C_2H_2/He$ (10:10:80, v/v/v), and $C_2H_6/C_2H_4/C_2H_2/He$ (9:9:2:80, v/v/v/v) gas mixtures, CPOC-301 powder was subjected to packing into a custom-made stainless-steel column (3.0 mm I.D. × 120 mm) having void space filled with silica wool. Activating sample was accomplished through heating the packed column at 100 °C for 12 h under a constant He flow (10 mL $min^{-1}$ at 298 K and 1 bar). The He flow was then turned off and the C2 hydrocarbon gas mixture was permitted to flow into the column (2 mL $min^{-1}$). The mass spectrometer was employed to continuously monitor the outlet effluent from column. After the breakthrough experiment, the sample was regenerated in situ in the column at 100 °C for 12 h. The complete breakthrough of $C_2H_6$ was determined using downstream gas composition reaching that of feed gas. On the basis of the mass balance, the gas adsorption capacities can be attained from the following equation[65]:

$$q_i = \frac{C_i V}{22.4 \times m} \times \int_0^t \left(1 - \frac{F}{F_0}\right) dt$$

where $q_i$ refers to the equilibrium adsorption capacity of gas $i$ (mmol $g^{-1}$), $C_i$ represents the feed gas concentration, $V$ refers to the volumetric feed flow rate (cm$^3$ $min^{-1}$), $t$ represents the adsorption time (min), $F_0$ and $F$, respectively, refer to the inlet and outlet gas molar flow rates, and $m$ represents adsorbent mass of (g). The separation factor ($\alpha$) of breakthrough experiment can be calculated as follows:

$$\alpha = \frac{q_A y_B}{q_B y_A}$$

In which $y_i$ is molar fraction of gas $i$ ($i = A, B$) in gas mixture.

**Binding energy calculations.** The molecular input of CPOC-301 was generated starting from its crystal structure, in which one of two disordered motifs was selected and the isobutyl group was reduced to a methyl group to simplify the simulation. The initial structure of CPOC-301 was then optimized by the semi-empirical extended tight-binding (xtb) program package developed by Grimme group[66]. The initial binding sites for $C_2H_6$, $C_2H_4$, and $C_2H_2$ were determined using noncovalent interaction (NCI)/iMTD algorithm in CREST according the procedure in the literature[59,60]. During binding site screening, CPOC-301 was free of any constraints, allowing structural relaxation as well as adaption to the C2 hydrocarbon guest. According to CREST calculation, a structure ensemble of NCI

complexes within a 6 kcal mol$^{-1}$ energy window is obtained, and the energetically lowest conformation was selected as the first binding site.

The free CPOC-301, C2 hydrocarbon guests, and C2@CPOC-301 were further optimized by DFT-D method by means of Dmol3 module as implemented in Accelrys Materials Studio package[67]. The PBE-type exchange-correlation functional with a generalized gradient approximation, the double numerical plus polarization basis sets that include a d-type polarization function on all non-hydrogen atoms and a p-type polarization function on all hydrogen atoms and Grimme method for DFT-D correction were employed for all calculations. Besides, FINE quality mesh size was employed in the calculations. The energy, force, and displacement convergence criteria were respectively set as $1 \times 10^{-5}$ Ha, $2 \times 10^{-3}$ Ha, and $5 \times 10^{-3}$ Å. The binding energies ($\Delta E$ bind in kJ mol$^{-1}$) were calculated as the differences in total energies $E$ between fully optimized C2@CPOC-301 and the CPOC-301 and C2 hydrocarbon guests in terms of the following equation:

$$\Delta E = E_{POC+gas} - E_{POC} - E_{gas}$$

where $E_{POC+gas}$ stands for the energy of the fully optimized C2@CPOC-301 structure, while $E_{POC}$ and $E_{gas}$ respectively represent energies of bare CPOC-301 structure and isolated C2 hydrocarbon molecule, respectively. Based on such an equation, more negative binding energy means more favorable binding. The detailed calculated results for energy of the C2@CPOC-301 are summarized in Supplementary Table 1.

## Data availability

All the experiment data that support the findings of this study are available within the article and Supplementary Information. Additional data are available from the corresponding author on reasonable request. The X-ray crystallographic data for CPOC-301 has been deposited at the Cambridge Crystallographic Data Centre (CCDC), under deposition number 1992563. This data file can be obtained free of charge from The Cambridge Crystallographic Data Centre via www.ccdc.cam.ac.uk/data_request/cif. Source data are provided with this paper.

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

## Acknowledgements

This work was financially supported by the National Key R&D Program of China (2017YFA0700102), the National Natural Science Foundation of China (22071244, 21707143), and the Strategic Priority Research Program of the Chinese Academy of Sciences (XDB20000000).

## Author contributions

DY and KS proposed the ideas and supervised the project. KS and SD synthesized the cages and conducted their characterization. WW and CJ performed gas sorption and separation experiments. DY, KS, and WW analyzed data and wrote the manuscript. All authors discussed the results and commented on the manuscript.

## Competing interests

The authors declare no competing interests.
