## [Peer Review File · Nature Communications]

REVIEWER COMMENTS

Reviewer #1 (Remarks to the Author):

The separation of olefin and paraffin is an important process in industry and the design of efficient adsorbents has attracted immense attention. Owing to the advantage of alkane-selective adsorbents, the related research has been explored. Here, the author designed a novel truncated octahedral calix[4]resorcinarene-based POC adsorbent (CPOC-301) with weakly polar pore surface, and it shown ethane-selective performance with fairly good capacity. The work revealed the first example of ethane-selective material in porous organic cages, which provided a novel way to the design of alkane-selective materials and the deep insight into the alkane-selective separation mechanisms. The experiment of the work is well organized. However, to reveal the actual adsorption behavior of the different C₂ hydrocarbon, the crystal structure with guest molecules is necessary, which is able to give deep understanding about the separation mechanisms and make the manuscript more meaningful. Therefore, further improvement is required for publication in NC. The other questions are as follows:

1. The desorption curve of N₂ gas sorption isotherm is strange. It should be coincided with the adsorption curve since there is no strong interaction. The results should be checked.
2. The unit of gas uptake in Q_{st} curve is better to be consistent with that in adsorption isotherms for easy understand.
3. The most intuitive evidence to verify stability of materials should be the N₂ gas sorption isotherm. The N₂ adsorption results as well as the PXRD of the sample after exposed to air should be supplied.
4. The breakthrough performance of C₂H₂/C₂H₄/C₂H₆ mixtures should be provided.

Reviewer #2 (Remarks to the Author):

This manuscript describes the application of a calixresorcinarene-based cage, previously reported by the authors, for ethane/ethene separation. The manuscript includes measurements of separation performance based on isotherms and IAST calculations and dynamic breakthrough experiments. The authors state that this is the first demonstration of ethane/ethene separation by an organic cage. The mechanism for the observed selectivity is investigated by molecular simulations.

The experimental characterization of the adsorption and breakthrough performance appear to be sound. However, the authors should provide some context for the selectivities, for example, by comparing the performance of other porous organic materials, such as hydrogen-bonded organic frameworks (e.g. 10.1021/jacs.9b12428), porous aromatic frameworks (10.1002/chem.201900857), etc. (review 10.1002/chem.201904455) Arguably, the authors statement that this manuscript reports "POC adsorbent for C₂H₄ purification for the first time" should be clarified, as Mastalerz et

al. previously reported a boronate ester cage with potential for C₂ hydrocarbon separations, albeit based on Henry and IAST calculated selectivities pairs of ethane, ethene and acetylene (DOI 10.1002/chem.201802123).

Although the experimental measurements seem to be robust and well described, the report of the molecular modelling used to explore the adsorption of the guest, and hence the origin of the selectivity is rather unclear. The main text states that DFT-d methods were used to determine the primary guest adsorption site. However, this cannot be the method used to initially locate these sites. The methods section suggests the initial sites were determined using a simulated annealing approach, but the detail is too sparse here and explaining the use of DMol3. There is no further information in the ESI. It is impossible to judge whether the methods used to determine the sites are appropriate and if they have been appropriately applied, and therefore whether the results shown are likely to be robust. The authors should provide a fuller description of the process for modelling the adsorption sites.

Notwithstanding uncertainty about the simulation methods, the authors propose that interactions between the guest hydrogen atoms and the arene are stronger for the C₂H₆ than C₂H₄ because it is more polarizable, but they then go on to say that it is because there are more C–H...π H-bonds. This section is quite confusing, partly because collectively it is true that the binding energy suggests stronger interaction between ethane and the host, but arguably if these H-bonds dominate, as they are more numerous, individually each C–H...π is weaker. The evidence here for the polarization theory is not particularly strong. The authors could consider looking at the electrostatic potential of the host and guest to explore this, for example, mapping on the Hirshfeld surface. They should also include any relevant references to studies of selectivity in other organic materials to strengthen their proposed mechanism. Currently, although it is intuitive, it is not fully supported by the results presented and the authors do not provide any literature to show it has been established previously. Although the system may not be at equilibrium or saturation, the authors should comment on whether additional adsorption sites are expected other than the six equivalent primary sites from the measured gas uptake, or whether all six sites are occupied.

In summary, the manuscript reports the application of an example of a relatively recent class of materials (POCs) to an industrially relevant separation. Although the experimental measurements of gas adsorption and breakthrough separations are satisfactory, the interpretation of the results in the context of the wider field are limited. It is unclear how the molecular simulations included were performed and this undermines the results. The interpretation is also somewhat superficial and confusing and this aspect also lacks reference to previous studies of these types of guests in porous organic hosts. Without major revision, particularly of the simulation sections, I would not recommend publication in Nature Communication. As an experimental study alone, it is slightly limited in scope, but could perhaps be augmented, for example, by measurements of other cages in the family.

Point by point response to the reviewers' comments

Dear Reviewers:

We wish to express our appreciation to the referees for their great efforts and suggestions for our manuscript. These comments are valuable and very helpful for our revisions and improvements to our paper. We have tried our best to improve the manuscript and made some changes in the manuscript and the supplementary information. We hope that the revised manuscript will be accepted.

Reviewer #1:

Comments:

The separation of olefin and paraffin is an important process in industry and the design of efficient adsorbents has attracted immense attention. Owing to the advantage of alkane-selective adsorbents, the related research has been explored. Here, the author designed a novel truncated octahedral calix[4]resorcinarene-based POC adsorbent (CPOC-301) with weakly polar pore surface, and it shown ethane-selective performance with fairly good capacity. The work revealed the first example of ethane-selective material in porous organic cages, which provided a novel way to the design of alkane-selective materials and the deep insight into the alkane-selective separation mechanisms. The experiment of the work is well organized. However, to reveal the actual adsorption behavior of the different C2 hydrocarbon, the crystal structure with guest molecules is necessary, which is able to give deep understanding about the separation mechanisms and make the manuscript more meaningful. Therefore, further improvement is required for publication in NC. The other questions are as follows:

Response:

We thank you for your positive comments on our work. We have tried our best to get the crystal structure with C2 hydrocarbon guest molecules by single-crystal X-ray diffraction. Although this method for structure analysis has been achieved in porous metal-organic framework (MOF) system, it was unsuccessful in our case due to the following two reasons: (1) the peak intensities of x-ray diffraction from **CPOC-301** are relatively weak, because there is no heavy atom in the molecular structure of **CPOC-301**. In fact, it is not able to locate the guest solvent molecules in the inner cavity of **CPOC-301**, even we have tried many times to collect the single crystals by increasing the exposure time; (2) the crystals of **CPOC-301** easily crack into pieces when removed from the mother liquor for a short time, and quickly turn into pieces or powder under vacuum, because the packings of these organic cages are via weak supramolecular interactions. Notably, the pieces or powder have been characterized by PXRD, H-NMR, mass spectrometry and gas sorption, which showed that **CPOC-301** is robust.

According to your suggestions:

Q1. The desorption curve of N₂ gas sorption isotherm is strange. It should be coincided with the adsorption curve since there is no strong interaction. The results should be checked.

Response:

Thanks for the useful suggestion and comments. We have repeated the desorption curve of N₂ gas sorption isotherm of **CPOC-301** several times, which all showed the same trend. In fact, all the desorption curve of nitrogen gas sorption isotherms of the reported calix[4]resorcinarene-based porous organic cages with high BET surface do not coincide with the adsorption curve (*J. Am. Chem. Soc.* **2020**, *142*, 18060-18072), and this phenomenon has been observed in other porous organic cages (see examples as *J. Am. Chem. Soc.* **2014**, *136*, 1438-1448; *Chem. Commun.* **2015**, *51*, 1976-1979; *Nat. Chem.* **2017**, *9*, 17-25; *Cryst Growth Des.* **2019**, *19*, 3647-3651; *Angew Chem Int Ed.* **2019**, *58*, 8819-8823; *Angew. Chem. Int. Ed.* **2020**, *59*, 19675-19679).

Q2. The unit of gas uptake in Qst curve is better to be consistent with that in adsorption isotherms for easy understand.

Response:

Thanks for this useful suggestion. We have changed the unit of gas uptake in Qst curve of the C2 hydrocarbons to be consistent with that in adsorption isotherms.

Q3. The most intuitive evidence to verify stability of materials should be the N₂ gas sorption isotherm. The N₂ adsorption results as well as the PXRD of the sample after exposed to air should be supplied.

Response:

Thanks for the useful suggestion and comments. We have added the N₂ adsorption results (**Figure C1**) as well as the PXRD of the sample after exposed to air (**Figure C1**) in the revised supporting information.

Figure C1. N₂ gas sorption isotherm at 77 K for **CPOC-301** after being exposed to air for a week.

Figure C2. PXRD of CPOC-301 after being exposed to air for a week.

Q4. The breakthrough performance of C₂H₂/C₂H₄/C₂H₆ mixtures should be provided.

Response:

Thanks for the useful suggestion and comments. We have added the breakthrough performance of C₂H₂/C₂H₄/C₂H₆ (1:1:1) ternary mixture of CPOC-301 (Figure C3), which further confirmed our experimental Q_{st} results (the host-guest interactions between CPOC-301 and C₂ hydrocarbon are in the order of C₂H₂<C₂H₄<C₂H₆).

Figure C3. Experimental breakthrough curves for C₂H₂/C₂H₄/C₂H₆ (1:1:1) ternary mixture at 298 K and 1 bar over a packed bed of CPOC-301.

Reviewer #2:*Comments:*

Q1. This manuscript describes the application of a calixresorcinarene-based cage, previously reported by the authors, for ethane/ethene separation. The manuscript includes measurements of separation performance based on isotherms and IAST calculations and dynamic breakthrough experiments. The authors state that this is the first demonstration of ethane/ethene separation by an organic cage. The mechanism for the observed selectivity is investigated by molecular simulations. The experimental characterization of the adsorption and breakthrough performance appear to be sound. However, the authors should provide some context for the selectivities, for example, by comparing the performance of other porous organic materials, such as hydrogen-bonded organic frameworks (e.g. 10.1021/jacs.9b12428), porous aromatic frameworks (10.1002/chem.201900857), etc. (review 10.1002/chem.201904455) Arguably, the authors statement that this manuscript reports "POC adsorbent for C₂H₄ purification for the first time" should be clarified, as Mastalerz et al. previously reported a boronate ester cage with potential for C₂ hydrocarbon separations, albeit based on Henry and IAST calculated selectivities pairs of ethane, ethene and acetylene (DOI 10.1002/chem.201802123).

Response:

Thanks for the useful suggestion and comments. We have added the context for comparing the selectivity of **CPOC-301** with other porous organic materials that prefer to adsorb C₂H₆ than C₂H₄ (including *J. Am. Chem. Soc.* **2020**, *142*, 633-640); *Angew. Chem. Int. Ed.* **2021**, *60*, DOI: 10.1002/anie.202100342). Notably, the reported porous aromatic frameworks in (10.1002/chem.201900857) and (10.1002/chem.201904455) all show the prefer to adsorb C₂H₄ than C₂H₆, which are different to **CPOC-301**. Thus, we do not compare the selectivity of **CPOC-301** with the abovementioned porous aromatic frameworks. Moreover, we have revised the ambiguous statement (POC adsorbent for C₂H₄ purification for the first time), compare the selectivity of **CPOC-301** with the boronate ester cages, and also cited the reference (DOI: 10.1002/chem.201802123) in the revised manuscript.

Q2. Although the experimental measurements seem to be robust and well described, the report of the molecular modelling used to explore the adsorption of the guest, and hence the origin of the selectivity is rather unclear. The main text states that DFT-d methods were used to determine the primary guest adsorption site. However, this cannot be the method used to initially locate these sites. The methods section suggests the initial sites were determined using a simulated annealing approach, but the detail is too sparse here and explaining the use of DMol3. There is no further information in the ESI. It is impossible to judge whether the methods use to determine the sites are appropriate and if they have

been appropriately applied, and therefore whether the results shown are likely to be robust. The authors should provide a fuller description of the process for modelling the adsorption sites.

Response:

Thanks for the useful suggestion and comments. We have recalculated the adsorption sites by referring to Professor Stefan Grimme's method which published in *J. Phys. Chem. C* (Spicher, S.; Bursch, M.; Grimme, S. Efficient Calculation of Small Molecule Binding in Metal-Organic Frameworks and Porous Organic Cages. *J. Phys. Chem. C* **2020**, *124*, 27529-27541). The molecular input of **CPOC-301** was generated starting from its crystal structure, in which one of two disordered linkers was selected and the isobutyl group was reduced to a methyl group for simplifying the simulation. And then the initial structure was optimized by the semiempirical extended tight-binding (xtb) program package developed by the Grimme group. The C2 molecules were placed manually at the center of mass of the **CPOC-301**. To screen for different binding sites, the noncovalent interaction (NCI)/iMTD algorithm in CREST is employed. **CPOC-301** was free of any constraints, allowing structural relaxation and adaption to the C2 guest. From the CREST calculation, a structure ensemble of NCI complexes within a 6 kcal mol⁻¹ energy window is obtained. The energetically lowest conformation was selected as the first binding site. The free **CPOC-301**, C2 guests and binding site C2@**CPOC-301** were further optimized by first-principles dispersion-corrected density functional theory (DFT-D) method by the Dmol3 module as implemented in the Accelrys Materials Studio package. The widely used generalized gradient approximation (GGA) with the Perdew-Burke-Ernzerhof (PBE) functional and the double numerical plus d-functions (DNP) basis set, Grimme method for DFT-D correction were used. The energy, force and displacement convergence criterions were set as 1 × 10⁻⁵ Ha, 2 × 10⁻³ Ha and 5 × 10⁻³ Å, respectively. Binding energies (ΔE_{bind} in kJ mol⁻¹) are calculated as the differences in total energies E between the fully optimized C2@**CPOC-301** and the free **CPOC-301** and C2 guests ($\Delta E = E_{\text{CPOC+gas}} - E_{\text{CPOC}} - E_{\text{gas}}$). In fact, the calculated binding sites are consistent with the calculated results last time, and the variation trend of the bonding energy is also consistent. We have added fuller description of the process for modelling the binding sites in the revised manuscript in binding energy calculations section as well as the calculated result in the revised supporting information (**Table C1**).

Table C1 The binding energies (ΔE) for C2@**CPOC-301** calculated by the Dmol3 module.

	$E_{\text{CPOC+gas}}$ (ha)	E_{CPOC} (ha)	E_{gas} (ha)	ΔE (ha)	ΔE (kJ/mol)
C₂H₆	-16112.1123535	-16032.3607175	-79.7353730	-0.0162630	-42.7
C₂H₄	-16110.8772222	-16032.3607175	-78.5008139	-0.0156908	-41.2
C₂H₂	-16109.6227364	-16032.3607175	-77.2480486	-0.0139703	-36.7

Q3. Notwithstanding uncertainty about the simulation methods, the authors propose that interactions between the guest hydrogen atoms and the arene are stronger for the C₂H₆ than C₂H₄ because it is more polarizable, but they then go onto say that it is because there are more C–H...π H-bonds. This section is quite confusing, partly because collectively it is true that the binding energy suggests stronger interaction between ethane and the host, but arguably if these H-bonds dominate, as they are more numerous, individually each C–H...π is weaker. The evidence here for the polarization theory is not particularly strong. The authors could consider looking at the electrostatic potential of the host and guest to explore this, for example, mapping on the Hirshfeld surface. They should also include any relevant references to studies of selectivity in other organic materials to strengthen their proposed mechanism. Currently, although it is intuitive, it is not fully supported by the results presented and the authors do not provide any literature to show it has been established previously. Although the system may not be at equilibrium or saturation, the authors should comment on whether additional adsorption sites are expected other than the six equivalent primary sites from the measured gas uptake, or whether all six sites are occupied.

Response:

Thanks for the useful suggestion and comments. We have revised "the interactions between the guest hydrogen atoms and the arene are stronger for the C₂H₆ than C₂H₄ because it is more polarizable". Moreover, we have also added the Hirshfeld surface analysis of C₂ hydrocarbon guests with the calix[4]resorcinarene cavity host to show their intermolecular interactions (**Figure C4**). Notably, the strong intermolecular interactions between the C₂ hydrocarbons and calix[4]resorcinarene cavity are indicated as bright red spots on the Hirshfeld surface (Supplementary Fig. 19), which are close to the abovementioned interaction regions of C–H...π bonds between the C₂ guests and calix[4]resorcinarene host. We have cited the related references of other organic materials by using the DFT simulation method to calculate the interactions between the C₂ gas molecules and framework host as well as their selectivity (see example as *J. Am. Chem. Soc.* **2020**, *142*, 633-640; *Angew. Chem. Int. Ed.* **2021**, *60*, DOI: 10.1002/anie.202100342). Moreover, it is suggested that one organic cage can absorb about 22.5 C₂H₆ and 19.5 C₂H₄ molecules calculated from the adsorption curve, thus the six primary adsorption sites of the **CPOC-301** are occupied by C₂ hydrocarbon guest molecules, and there are additional adsorption sites. We have added this in the revised manuscript.

Figure C4. The Hirshfeld surface showing the intermolecular interactions of **a** C₂H₆, **b** C₂H₄ and **c** C₂H₂ with the cavities of calix[4]resorcinarene. Blue-white-red, which corresponds to electron density varying from 0.0 to 0.015 a.u.

Q4. In summary, the manuscript reports the application of an example of a relatively recent class of materials (POCs) to an industrially relevant separation. Although the experimental measurements of gas adsorption and breakthrough separations are satisfactory, the interpretation of the results in the context of the wider field are limited. It is unclear how the molecular simulations included were performed and this undermines the results. The interpretation is also somewhat superficial and confusing and this aspect also lacks reference to previous studies of these types of guests in porous organic hosts. Without major revision, particularly of the simulation sections, I would not recommend publication in Nature Communication. As an experimental study alone, it is slightly limited in scope, but could perhaps be augmented, for example, by measurements of other cages in the family.

Response:

Thanks for the useful suggestion and comments. We have added the C₂H₆ and C₂H₄ gas sorption curves and experimental column breakthrough results (**Figures C5-C8**) of other cages in the family including one hexameric octahedra with functional methyl group (**CPOC-301-Me**) and a trimeric triangular prism (**CPOC-201**). All the aforementioned three organic cages show the preferential adsorption with C₂H₆ over C₂H₄, and can also directly separate C₂H₄ from the C₂H₄/C₂H₆ mixture by column breakthrough experiments. These data further suggest that the C–H⋯π interactions existing between the C₂H₆ and calix[4]resorcinarene cavity are stronger than those of the C₂H₄ molecule. Moreover, the host-guest interactions between **CPOC-301** and C₂ hydrocarbon are in the order of C₂H₂<C₂H₄<C₂H₆, which have also been validated by the breakthrough performance of C₂H₂/C₂H₄/C₂H₆ mixtures of **CPOC-301** (**Figure C3**).

Figure C5. Experimental C_2H_6 and C_2H_4 adsorption isotherms of **CPOC-301-Me** at 293 K. Inset is the simulated molecule structure of **CPOC-301-Me**.

Figure C6. Experimental C_2H_6 and C_2H_4 adsorption isotherms of **CPOC-201** at 293 K. Inset is the molecule structure of **CPOC-201**.

Figure C7. The experimental column breakthrough curve of **CPOC-301-Me** with an equimolar C_2H_6/C_2H_4 mixture at 298 K.

Figure C8. The experimental column breakthrough curve of **CPOC-201** with an equimolar C_2H_6/C_2H_4 mixture at 298 K.

REVIEWERS' COMMENTS

Reviewer #1 (Remarks to the Author):

The detailed modeling studies and experiments have been conducted in the revised manuscript and its quality has been significantly improved. I suggest acceptance for publication after addressing following issues.

1. The C₂H₂ adsorption curve and C₂H₆/C₂H₂ selectivity should be provided.
2. The water stability of POCs is needed.
3. The C₂H₆/C₂H₄ separation performance comparisons of this work and the reported materials is needed to be included in the form of figure or table.
4. The author added some other POCs. The PXRD pattern comparisons of the synthesized samples and the simulated structures is needed. The C₂H₆/C₂H₄ selectivity should be provided.

Reviewer #2 (Remarks to the Author):

I appreciate the authors efforts to address the comments made during the previous review. I am satisfied that they have revised their manuscript to improve on the issues raised by both reviewers. I find that more expansive and rigorous description and discussion of the molecular simulations makes the interpretation much stronger, and strengthens the paper overall; I hope that the authors will agree, despite the additional work! I support accepting the revised paper into Nature Communications.

Point by point response to the reviewers' comments

Reviewer #1:

Comments:

The detailed modeling studies and experiments have been conducted in the revised manuscript and its quality has been significantly improved. I suggest acceptance for publication after addressing following issues.

Response:

We are thankful to you for accepting the revised manuscript and recommending for publication.

Q1. The C_2H_2 adsorption curve and C_2H_6/C_2H_2 selectivity should be provided.

Response:

Thanks for this useful suggestion and comment. We have added the C_2H_2 adsorption (Fig. S1) curve and C_2H_6/C_2H_2 selectivity (Fig. S2) in the revised supporting information.

Fig. S1 C_2H_2 adsorption/desorption isotherm of **CPOC-301** at 273, 283 and 293 K.

Fig. S2 Selectivity of **CPOC-301** predicted by the IAST method for an equimolar C_2H_6/C_2H_2 mixture at 293 K.

Q2. The water stability of POCs is needed.

Response:

Thanks for this useful suggestion and comment. We have added the water stability of **CPOC-301** (Fig. S3) in the revised supporting information.

Fig. S3 PXRD patterns of **CPOC-301** after being soaked in water for 24 hours.

Q3. The C_2H_6/C_2H_4 separation performance comparisons of this work and the reported materials is needed to be included in the form of figure or table.

Response:

Thanks for this useful suggestion and comment. We have added the C_2H_6/C_2H_4 separation performance comparisons of **CPOC-301** and the reported materials (Fig. S4) in the revised supporting information.

Fig. S4 The reported C_2H_6 -selective porous organic molecular materials, and several selected C_2H_6 -selective porous framework materials^{S1-S9}. Note: the C_2H_6 Uptake of boronic ester cage was measured at 273 K, and its actual C_2H_6/C_2H_4 separation performance by breakthrough experiment were not investigated.

Q4. The author added some other POCs. The PXRD pattern comparisons of the synthesized samples and the simulated structures is needed. The C_2H_6/C_2H_4 selectivity should be provided.

Response:

Thanks for this useful suggestion and comment. We have added the PXRD patterns (Figs. S5 and 6) and C_2H_6/C_2H_4 selectivity (1.18 and 1.20 in Supplementary Figs. 26 and 27, respectively) of **CPOC-301-Me** and **CPOC-201** in the revised supporting information.

Fig. S5 PXRD patterns of **CPOC-301-Me** after desolvation. This suggests that **CPOC-301-Me** is isostructural to **CPOC-301**.

Fig. S6 PXRD patterns of **CPOC-201** after desolvation.

References

- S1. Elbert S. M., *et al.* Shape-persistent tetrahedral 4+6 boronic ester cages with different degrees of fluoride substitution. *Chem. Eur. J.* **24**, 11438-11443 (2018).
- S2. Zhang X., *et al.* Selective ethane/ethylene separation in a robust microporous hydrogen-bonded organic framework. *J. Am. Chem. Soc.* **142**, 633-640 (2020).
- S3. Chen B., *et al.* A rod-packing hydrogen-bonded organic framework with suitable pore confinement for benchmark ethane/ethylene separation. *Angew. Chem. Int. Ed.* **60**, 10304-10310(2021).
- S4. He C. H., *et al.* Microregulation of pore channels in covalent-organic frameworks used for the selective and efficient separation of ethane. *ACS Appl. Mat. Interfaces.* **12**, 52819-52825 (2020).
- S5. Li L., *et al.* Ethane/ethylene separation in a metal-organic framework with iron-peroxo sites. *Science.* **362**, 443-446 (2018).
- S6. Qazvini O. T., Babarao R., Shi Z.-L., Zhang Y.-B. & Telfer S. G. A robust ethane-trapping metal-organic framework with a high capacity for ethylene purification. *J. Am. Chem. Soc.* **141**, 5014-5020 (2019).
- S7. Lin R.-B., *et al.* Boosting ethane/ethylene separation within isoreticular ultramicroporous metal-organic frameworks. *J. Am. Chem. Soc.* **140**, 12940-12946 (2018).
- S8. Zeng H., *et al.* Cage-interconnected metal-organic framework with tailored apertures for efficient C₂H₆/C₂H₄ separation under humid conditions. *J. Am. Chem. Soc.* **141**, 20390-20396 (2019).
- S9. Liao P.-Q., Zhang W.-X., Zhang J.-P. & Chen X.-M. Efficient purification of ethene by an ethane-trapping metal-organic framework. *Nat. Commun.* **6**, 9697 (2015).